behaviour/bioengineering/ecology

collective building, behavioural plasticity, *Acromyrmex fracticornis*, quantitative stigmergy, self-organization, leaf-cutting ants

**Author for correspondence:**
Daniela Römer
e-mail: daniela.roemer@uni-wuerzburg.de

# Selection and spatial arrangement of building materials during the construction of nest turrets by grass-cutting ants

Daniela Römer[1,2], Marcela I. Cosarinsky[3] and Flavio Roces[1]

[1]Department of Behavioural Physiology and Sociobiology, Biocenter, University of Würzburg, Am Hubland, 97074 Würzburg, Germany
[2]Unidad de Entomología, Departamento de Protección Vegetal, Facultad de Agronomía, Universidad de la República, Avenue E. Garzón 780, CP 12900 Montevideo, Uruguay
[3]Departamento de Ciencias Geológicas, Facultad de Ciencias Exactas y Naturales, Universidad de Buenos Aires, Pabellón II, Ciudad Universitaria, Buenos Aires, Argentina

DR, 0000-0002-7437-2195; MIC, 0000-0003-1535-3469;
FR, 0000-0001-9258-3079

Ants build complex nest structures by reacting to simple, local stimuli. While underground nests result from the space generated by digging, some leaf- and grass-cutting ants also construct conspicuous aboveground turrets around nest openings. We investigated whether the selection of specific building materials occurs during turret construction in *Acromyrmex fracticornis* grass-cutting ants, and asked whether single building decisions at the beginning can modify the final turret architecture. To quantify workers' material selection, the original nest turret was removed and a choice between two artificial building materials, thin and thick sticks, was offered for rebuilding. Workers preferred thick sticks at the very beginning of turret construction, showed varying preferences thereafter, and changed to prefer thin sticks for the upper, final part of the turret, indicating that they selected different building materials over time to create a stable structure. The impact of a single building choice on turret architecture was evaluated by placing artificial beams that divided a colony's nest entrance at the beginning of turret rebuilding. Splitting the nest entrance led to the self-organized construction of turrets with branched galleries ending in multiple openings, showing that the spatial location of a single building material can strongly influence turret morphology.

# 1. Introduction

Constructing a dwelling is an ability shared by a variety of animals, like mammals [1], birds [2] and insects [3]. While these constructions may vary greatly in size and shape, they all provide a barrier to the environment [4], thus protecting the inhabitants from predation or fluctuation of environmental variables, allowing food storage and, in last consequence, the successful rearing of offspring. The built structures are species-specific and often described as an extended phenotype [5]. Aside from humans, social insects, especially ants, create some of the most complex nest structures in the animal kingdom. But how do ants manage to collectively build such structures?

A single ant worker appears not to be able to create a complex nest as it only displays rather simple behaviours. However, by means of a combination of different mechanisms, namely templates, stigmergy and self-organization, workers are able to coordinate their activities and build a complex structure adapted to a colony's needs [6,7]. Workers use only local information, and have no overview about the complete structure [8]. Ant workers are known to use environmental cues and the presence of brood or symbiotic partners as templates that spatially guide the building process [9,10], so that the resulting nest is adapted to its environment [11,12]. During self-organized collective building, workers react to each other and the interactions influence the collective response directly through positive or negative feedback loops. For instance, positive feedback and vibrational communication [13] draw building activity to certain sites, concentrating and coordinating collective building, while a negative feedback as a result of the building activity leads to the cessation of building [8,14]. Through stigmergy, ant workers do not even have to communicate directly to coordinate their building behaviour. Rather, the modification of the environment by one building worker acts as a stimulating configuration, triggering further building behaviour at this spot, either by the same individual or other workers [15,16].

Most ants build their nests underground by digging into the soil to create space in the shape of tunnels and chambers. Workers transport the excavated soil outside and dispose of it forming a mound of variable height above the underground nest structure, which may protect the colony against flooding and help to ventilate the nest, for example in leaf-cutting ants. Colonies of *Atta* leaf-cutting ants excavate up to 8000 chambers to maintain their fungus gardens, and nests can reach 7 m in depth [17–19], so that sufficient gas exchange may be compromised. In *Atta vollenweideri*, the shape of the nest mound promotes the ventilation of the underground chambers via a passive mechanism driven by wind [20].

Colonies of a number of leaf-cutting ant species not only build mounds above their underground nest, they also assemble special structures, called turrets, on top of the mounds, surrounding the nest openings. Turrets promote the passive ventilation of the nest by increasing the height differences between the uppermost and the lowest nest openings on the mound [21,22]. The largest turrets are constructed by the grass-cutting ant *At. vollenweideri* [20,23], which inhabits clay-rich and often flooded soils. Turrets can reach a height of 15 cm and are composed of soil pellets originated from underground digging and plant fragments collected from the environment aboveground. The ants erect the turret walls by arranging the transported soil pellets around a nest opening, and by incorporating plant fragments into welded soil pellets, as revealed by microstructural analyses [24]. Nest turrets of other grass-cutting ant species (*Acromyrmex landolti*, *Acromyrmex balzani* and *Acromyrmex fracticornis*), on the other hand, are rather small. They average 3 cm in height [25], and mostly consist of dried grass blades arranged into a chimney-shaped, net-like structure [26,27] to form a single gallery, plastered with soil pellets, generally ending in one opening at the top, although turrets with several openings can sometimes be observed (figure 1). The adaptive value of these turrets is less understood. Protection against flooding or use as a visual homing cue have been discussed as main functions [26,28].

It is still an open question whether ants show material selection during the construction of such a structure or just randomly pile up materials transported from the nest or found in the nest vicinity. When observing the turret construction of *At. vollenweideri* in the laboratory, experiments showed that workers were not selective in their choice between two different building materials, i.e. small or large sand grains [29]. However, they carefully chose the location in the turret where they incorporated the different materials, which probably helped to build a porous yet mechanically stable structure, as shown for the assembly of walls in the rock-cavity dwelling ant *Temnothorax albipennis*. In this species, workers did show material selection when they build the nest wall, using differently sized sand grains to surround workers, queen and brood, effectively creating a simple nest [30]. Workers appeared to assess their building materials and showed individual preferences that change over time and with transportation distance [31].

Recently, a micromorphological study on the nest turrets of the grass-cutting ant *Ac. fracticornis* indicated that workers do not simply pile up available building materials indiscriminately around the

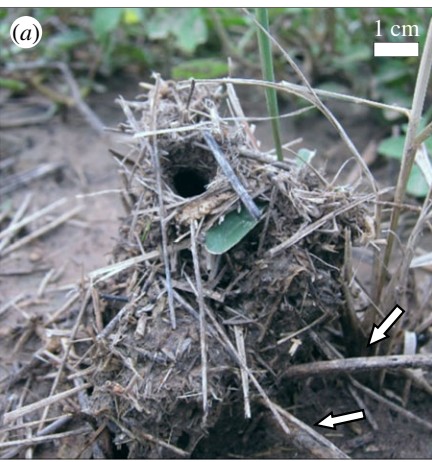
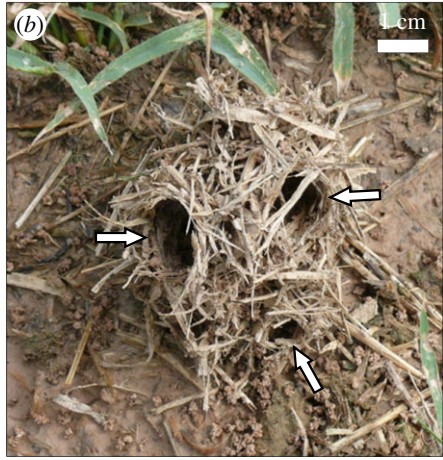

**Figure 1.** General morphology of natural turrets of *Acromyrmex fracticornis*. (*a*) Turret with a single opening. White arrows indicate two visible thick, 'beams' incorporated at the turret base among the plant materials. (*b*) Turret with three openings (white arrows). Photo credits: (*a*): M. Bollazzi; (*b*) D. Römer.

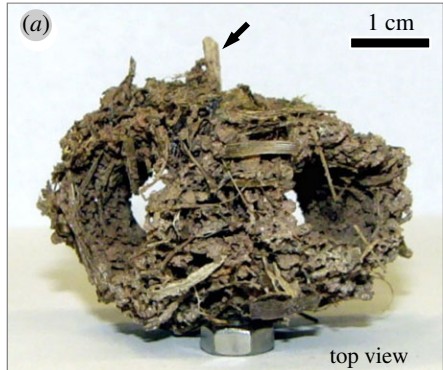
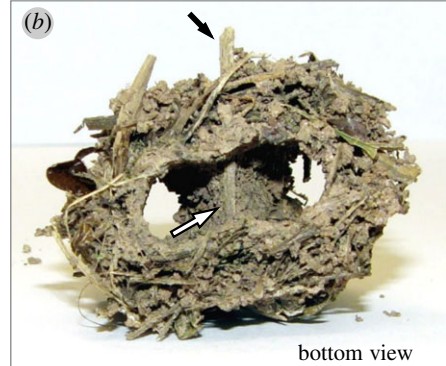

**Figure 2.** Turret with two nest openings built with natural materials. (*a*) Top view: black arrow indicates a visible beam protruding from the lower part of the turret. (*b*) Bottom view: the protruding beam (black arrow) bisected the turret gallery immediately above the single nest entrance (white arrow), leading to a tunnel bifurcation that ended in two separate openings at the top. Photo credits: F. Roces.

nest entrance. Rather, workers selected specific materials and intermesh thinner plant fragments with thicker sticks, cement the resulting plant mesh with soil pellets, and finally line the inner turret gallery with soil pellets [25]. More thick plant material is incorporated into the lower part of the turret, while the upper part is built mostly with thinner material. Based on the above descriptions [25], it is unclear whether such distinct spatial distribution of building materials is brought about by their availability at the different stages of the building process, or whether workers have preferences for specific building materials that change over time as the turret structure develops.

To investigate whether *Ac. fracticornis* workers do indeed show preferences for building materials that in addition change over time, we performed experiments on material selection in the field, aimed at providing a detailed insight into the behavioural mechanisms underlying the collective construction of nest turrets in ants. In a first experimental series, we offered equal numbers of artificial thin and thick sticks close to an *Ac. fracticornis* nest entrance after having removed the original turret, and recorded workers' material choices over time during the rebuilding of the turret. To understand the occurrence of turrets with branched galleries and more than one opening, as sometimes observed in the field (figure 1), we asked whether a single building decision at the beginning of the building process might lead to large changes in the final turret architecture. Visual examination of turrets with several openings often evinced the occurrence of a single plant fragment located across the nest opening, at the beginning of the gallery bifurcation (figure 2). This suggests that the choice of a single worker where to place a fragment during the construction process could strongly influence the final turret architecture, probably acting as base for further material placement. This hypothesis was explored in a

second experimental series, where we placed artificial beams across the exposed nest entrance of a colony after removal of the original turret, and analysed their effect on the number of openings and final architecture of the rebuilt turret.

# 2. Material and methods

## 2.1. Animals and study area

Experiments were performed with field colonies of the grass-cutting ant *Ac. fracticornis*, found in great abundance at the Reserva Ecológica 'El Bagual' (26°18′18.4″ S—57°49′51.0″ W) in the humid, eastern Chaco Region of Argentina, during two rainy seasons (February to March 2015 and March to April 2017). The field station is privately owned by the Estancia EL BAGUAL-ALPARAMIS S.A., and research was conducted with the permission of the owner Pablo Götz and the station supervisor Alejandro G. Di Giacomo. The study site was along dirt roads close to the field station in shallow grass that was either periodically mowed or grazed on by cattle. The soil in which *Ac. fracticornis* nests were located was an entic Hapludol (Mollisol), with an A horizon between 0 and 30 cm deep, an AC horizon at a depth between 30 and 60 cm underground, and a C horizon at 60–100 cm depth [25]. The species is not protected under the convention of international trade in endangered species (CITES).

## 2.2. Selection of building materials for turret construction

To quantify material selection by *Ac. fracticornis* workers, we performed controlled field experiments by offering a choice of two building materials (thin and thick sticks) for reconstruction after removal of the original nest turret. Workers' material preferences were quantified in two subsequent experimental phases over 1.5 h each. Observations during the first phase, called 'initial building phase', were made immediately after the removal of the original turret in the afternoon, starting around the time 17.00, when both temperature and solar radiation decreased, and visible worker activity increased. Foraging activity was completely nocturnal in these colonies, because of the very high daily temperatures. The second phase, called 'subsequent building phase', started the next morning after sunrise around 6.00. Worker activity visibly decreased with the increase in both temperature and solar radiation over time of day. At the beginning of each phase, we offered 60 artificial sticks (30 thick and 30 thin) for workers to choose from and recorded their choices by direct observations.

On the day of the experiment, all small plants growing around the turret of a selected *Ac. fracticornis* colony in a diameter of 30 cm were first removed with a small gardening shovel. Then the ground was cleared of any plant material that could potentially be used for turret construction, using a large paint brush. In this cleared space, 30 thin sticks (cut to length out of dry grass blades found around the station; mass: $4.7 \pm 1.5$ mg, length: $23.9 \pm 1.5$ mm, thickness: $0.5 \pm 0.2$ mm, $n = 30$) and 30 thick sticks (made from wooden tongue depressors, cut into smaller pieces, mass: $35.6 \pm 8.1$ mg, length: $22.2 \pm 1.7$ mm, width: $2.4 \pm 0.5$ mm, thickness: $1.7 \pm 0.1$ mm, $n = 30$) were placed in a circle around the nest turret, at a distance of 10 cm from the nest entrance (figure 3*a*). The stick length was chosen based on observations of natural materials carried by workers in the field. To better distinguish between thin and thick sticks during observations, thin sticks were dyed green by briefly shaking them in a container with paint pigment powder (Syria Green, Malzeit Künstlerpigmente, Germany). Previous observations in the laboratory indicated that grass-cutting ants do not show avoidance or preference for building materials coloured with this dye (*At. vollenweideri*, electronic supplementary material, figure S1). The original nest turret was removed at its base with a shovel and the opening of the nest slightly sprayed with water, using a spray bottle, to trigger rebuilding. During the next 1.5 h before sunset, i.e. during the 'initial building phase', the pick-up choices of workers selecting either a thin or a thick stick for building were recorded. They included sticks deposited closely to or directly on top of the nest entrance, and those carried into the nest that were subsequently carried back out and deposited. During this time period, only a small fraction of the total sticks offered was collected, so that no effect of stick availability on workers' choices was expected during the observation period. No observations were made during the night, although the process of turret rebuilding with natural materials continued overnight. Pick-up choices were again scored the following morning in the 'subsequent building phase', starting after sunrise. All leftover sticks from the previous initial building phase were removed, and 60 new sticks (30 thin and 30 thick) were placed around the nest opening as in the previous phase.

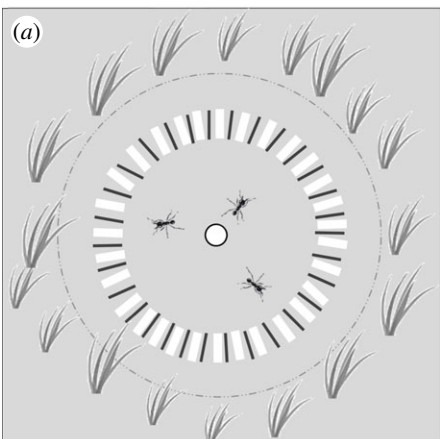 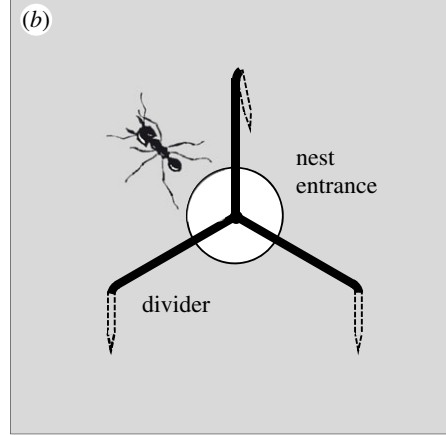

**Figure 3.** Experimental set-up. (*a*) Choice experiment: choice between thick (white) and thin (black) sticks as building material, placed 10 cm away from the nest entrance after the original nest turret was removed. An area of 30 cm diameter, indicated by the dashed line, was cleared of vegetation and potential natural building materials. The nest entrance is depicted as a white circle. Please note that the drawing is not to scale. (*b*) Divider experiment: the metal divider (arm length 3 cm) was placed on top of the exposed nest opening so as to divide the entrance in three similar sectors, and secured in the earth with prongs. Ant drawing by Griselda Roces.

A total of 24 experiments comprising the two building phases were performed with independent colonies. Not all of these 24 tested colonies showed building responses. In the initial phase, 21 out of 24, and in the subsequent phase, 19 out of 24, showed building activity. Building material choice was statistically analysed using SigmaPlot 11 (Systat Software, Inc.). After testing for normality (Shapiro–Wilk test), a Mann–Whitney rank sum test was performed. Count data was analysed using either the G-test for the goodness of fit to the ratio 1 : 1, or Fisher's exact test.

## 2.3. Spatial location of single building materials as determinants of final turret architecture

We tested whether the building decisions of single individuals (here, the placement of material across the nest opening or turret gallery) influence the final turret architecture. In this experimental series, we selected 40 nests with a turret having a single gallery and opening. The original turret was then removed at its base with a garden shovel. Twenty nests were randomly chosen to serve as control, and workers could rebuild the nest turret with natural materials from the surroundings without any experimental manipulation. Across the exposed nest opening of the remaining 20 experimental colonies, a three-armed metal divider splitting the entrance (figure 3*b*) was placed, and workers could also rebuild the nest turret as in the control group. After 24 h, the number of openings of the rebuilt turrets was counted for each nest. While a complete turret construction may last around 3 days, our previous work showed that most turrets reached their final external size after 1 day, with only a few slightly growing during the second day [25]. In total, 19 control and 16 experimental turrets were scored; the remaining turrets were found damaged and therefore not considered. The effect of divider placement on turret architecture was analysed using Fisher's exact test.

# 3. Results

## 3.1. Selection of building materials for turret construction

As soon as the original turret was removed, there was a visible increase in worker activity around the nest entrance and, in some experiments, the first offered sticks were picked up within a few minutes after the assays started. Workers usually did not put down a selected stick. Rather, they spent considerable time carrying or sometimes dragging, especially when small workers selected a thick stick, the selected material towards the nest entrance. Workers placed the first materials in the initial phase rather disorderly, just covering the nest opening. When we returned the next morning for testing in the subsequent phase, some materials appeared to have been moved, now laying tangentially around the nest opening, while turret height had notably increased.

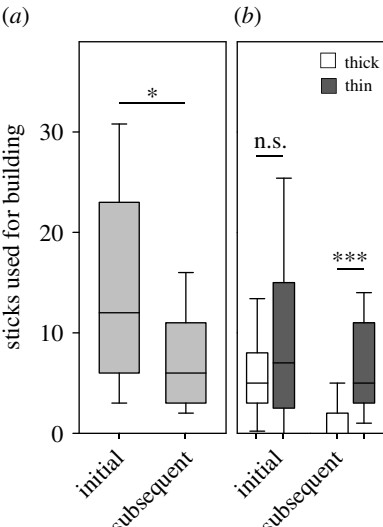

**Figure 4.** Comparison of chosen building materials (thick and thin sticks) between the initial building phase in the evening, after removal of the original turret and the subsequent building phase the next morning. (*a*) Total number of sticks (thick plus thin) used for building. (*b*) Data plotted separately for thick (white bars) and thin (dark grey bars) sticks. Mann–Whitney rank sum test, line: median, box: 25–75% percentiles, whiskers: min/max values without outliers. n.s., not significant, $^*p \leq 0.05$, $^{***}p < 0.001$.

Building activity, expressed as the total number of thick and thin sticks selected and incorporated in the turret, was significantly higher in the initial building phase than in the subsequent one (figure 4*a*; Mann–Whitney rank sum test, $U = 118.5$, $n_{initial} = 21$, $n_{subsequent} = 19$, $p = 0.029$), with a median of 12 sticks per colony being selected for building in the initial phase (25% percentile = 6, 75% percentile = 23), and six sticks in the subsequent phase (25% percentile = 3, 75% percentile = 11). Workers chose different building materials depending on phase of the turret construction. In the initial phase, when workers started to rebuild the turret, they overall built with a similar number of thick and thin sticks (figure 4*b*; Mann–Whitney rank sum test, $U = 175$, $n_{thick} = 21$, $n_{thin} = 21$, $p = 0.26$), while in the subsequent phase the next morning, they significantly preferred thin building material (Mann–Whitney rank sum test, $U = 43$, $n_{thick} = 19$, $n_{thin} = 19$, $p < 0.001$). A video of an ant picking up a thick stick and placing it on the nest opening can be found in the electronic supplementary material, Video S1.

We asked whether the selection of the very first stick in each of the building phases was random, or it depended on the building phase considered, and therefore evaluated what type of stick was picked up first in the two phases. Figure 5*a* shows a worker picking up a thick stick at the very beginning of turret rebuilding. Despite showing no preferences for thick and thin material in the initial building phase overall, the first choice of stick favoured thick ones (figure 5*b*; 15 of 21 colonies, goodness of fit test, $p = 0.046$). In the subsequent building phase the next morning, workers showed a significant preference for a thin stick as the first stick to be incorporated (figure 5*b*; 16 of 19 colonies, goodness of fit test, $p = 0.002$). Therefore, the worker's choice for the first building material being selected differed between the two phases (figure 5*b*; Fisher's exact test, two-tailed, $p = 0.0005$).

For a more detailed insight into the variation of material choice over the building time, we analysed the choices of successive sticks during the two building phases (figure 6*a*). While the very first stick for turret rebuilding was a thick one during initial building, as indicated above, from the second to the 16th stick workers showed no preferences for any of the stick types (figure 6*a*; initial phase, G-test, for detailed statistical analysis see the electronic supplementary material, table S1), followed by an almost exclusive building with thin material (17th to 28th stick). Workers continued showing a statistically significant preference for thin material the next morning, in the subsequent building phase (figure 6*a*; G-test, for detailed statistical analysis see the electronic supplementary material, table S1), although some thick sticks were also used for building. In addition, there was a notable reduction of building activity over time in both phases. For exemplary photos of building in the initial and subsequent phase, see figure 6*b*–*d*.

The number of colonies showing building activity decreased with time. At the end of the observation period of each phase, workers from only five and six colonies participated, respectively. While some colonies were still building with the offered sticks afterwards, the number of participating colonies was too low for any statistical analysis beyond the selection of the 28th stick in the initial phase, and the 11th stick in the subsequent phase, and therefore not included in the analysis.

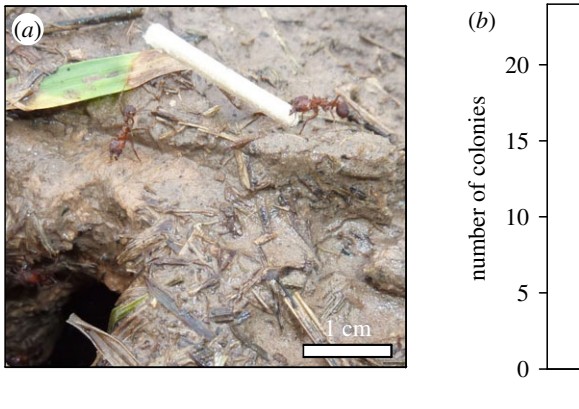

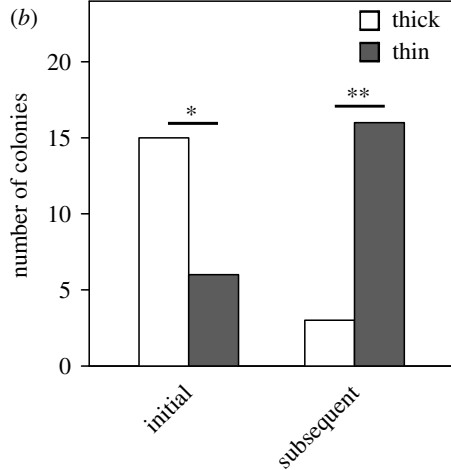

**Figure 5.** Choice of the very first stick for building. (*a*) A worker of *Acromyrmex fracticornis* selects a thick stick and carries it to the exposed nest entrance in the initial building phase, after removal of the original turret. (*b*) Number of colonies that selected either thick or thin sticks as the very first choice, for both the initial and the subsequent building phases. White bars, thick sticks, grey bars, thin sticks. G-test for goodness of fit to the ratio 1 : 1, $^*p \leq 0.05$, $^{**}p < 0.01$. Photo credits: D. Römer.

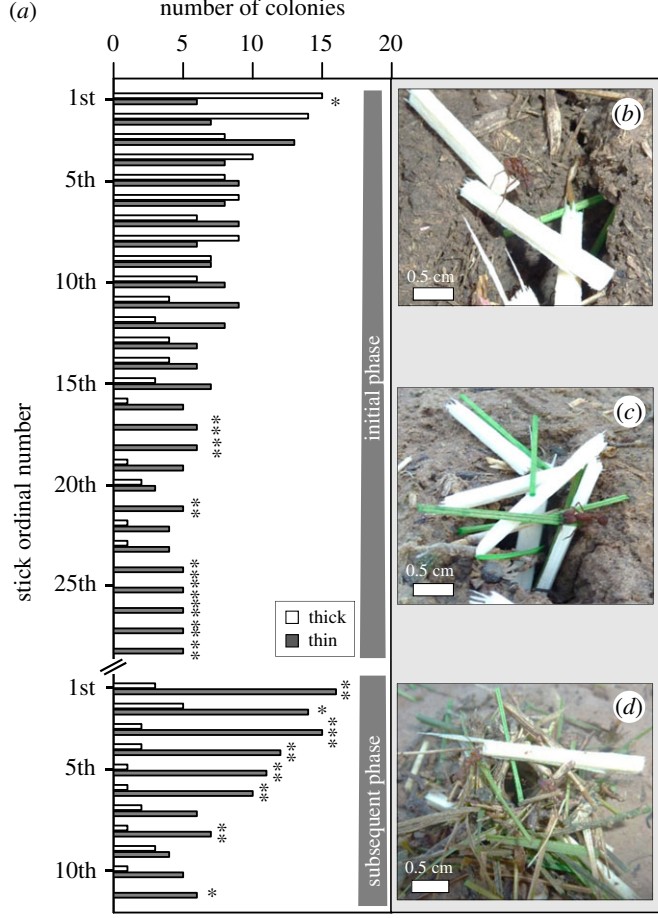

**Figure 6.** Choice of building materials over the two investigated building phases, each lasting 1.5 h. (*a*) Choice of consecutive sticks: white bars, thick sticks; grey bars, thin sticks. The *Y*-axis indicates the number of colonies that selected either thick or thin sticks over the two building phases; the *X*-axis depicts the ordinal stick number; G-test for goodness of fit to the ratio 1 : 1, $^*p \leq 0.05$, $^{**}p < 0.01$. (*b*) At the beginning of the rebuilding process, a worker is seen placing a thick stick near the nest entrance. (*c*) A worker is placing a thin stick across the nest entrance during the initial building phase. (*d*) Next morning, a worker is placing a thick stick on the nest turret during the subsequent building phase. Photo credits: D. Römer.

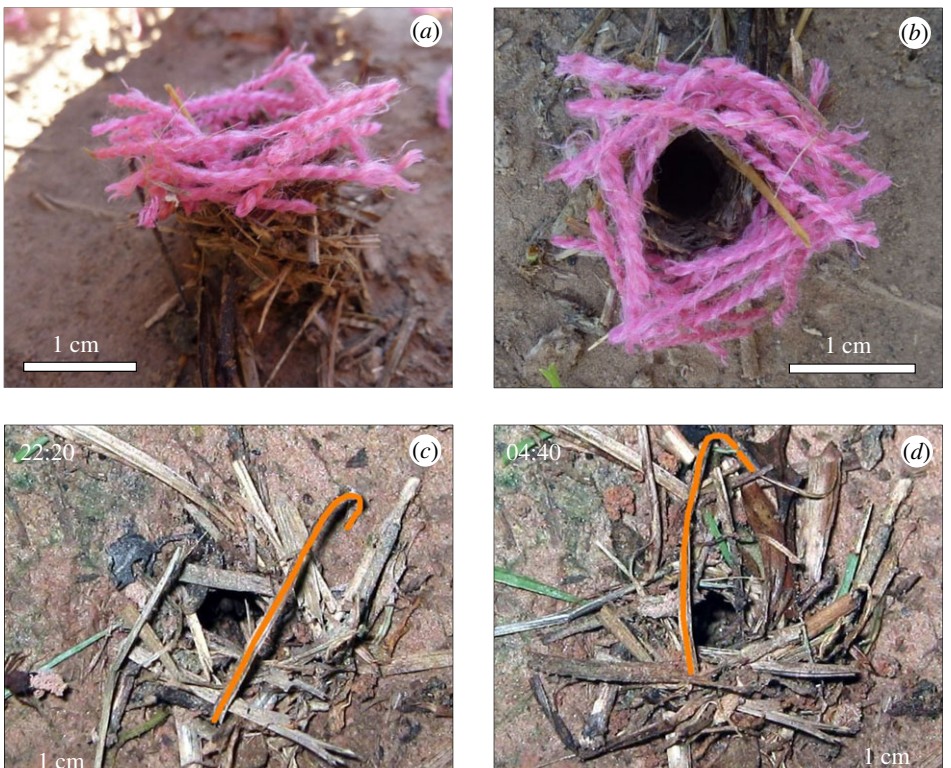

**Figure 7.** Examples of both tangential placement of building materials around the nest entrance and material displacement during turret building. (*a*) Side view and (*b*) top view of a newly built nest turret after removal of the original one. Pink wool pieces of the size of natural grass fragments were offered around the nest as building materials to better visualize their tangential piling and the overlap of the pieces' ends. (*c,d*) Shifting of a tangentially placed plant fragment from the right side of the turret opening to the left side during natural building (fragment marked in orange on the picture for visualization; photos taken at the indicated times, from Video 2, electronic supplementary material). Photo credits: D. Römer.

Even though the first sticks were usually placed across the nest opening, as seen in the pictures presented in figure 6*b,c*, the building materials were often placed at the edges and tangentially to the opening as the turret developed. The sticks were laid across each other at the corners, so that their ends overlapped, like in a human log cabin (figure 7*a,b*). Some material was laterally displaced around the turret at the initial building phase (figure 7*c,d*), which remained at a permanent position after additional material was added to consolidate the turret structure. An example of the natural rebuilding of a turret after removal of the original one, using both thick and thin plant materials, can be seen in the electronic supplementary material, Video S2.

## 3.2. Spatial location of single building materials as determinants of final turret architecture

After the removal of the original turret with one opening, workers constructed a new turret within 24 h in both the control and experimental series. The artificial splitting of the nest opening resulted in a significant higher number of rebuilt turrets with more than one opening (figure 8*a*; Fisher's exact test, $p = 0.0006$). From 19 rebuilt turrets of the control experiment, only 1 showed more than one opening, while of 16 rebuilt turrets with the divider placed across the nest entrance, 10 were reconstructed with two or more openings (for exemplary photos of rebuilt turrets see figure 8*b,c*). An example of a natural construction of a turret with more than one opening can be seen in Video S3 in the electronic supplementary material.

## 4. Discussion

Our study shows that turret building is not simply a process of ant workers haphazardly piling up building material on top of each other around the nest entrance. Rather, workers showed changing material selection over time. They preferentially incorporated thick sticks at the base of the

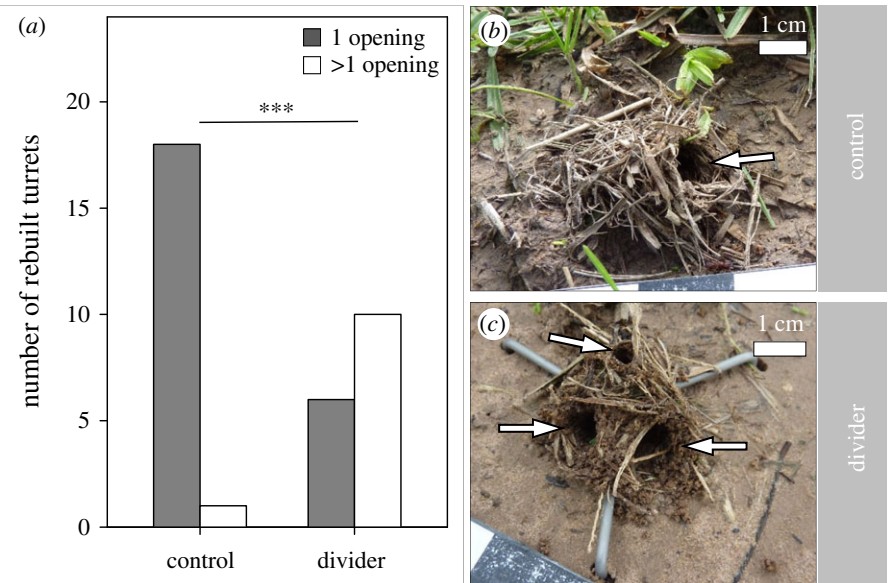

**Figure 8.** Divider experiment. (*a*) Number of rebuilt turrets with one or more than one opening in both the control and the divider experiment. Grey bars, turrets with one opening; white bars, turrets with more than one opening. Fisher's exact test, ***$p < 0.001$. (*b*) A rebuilt turret with a single opening (control). (*c*) A rebuilt turret with three openings (divider experiment). Turret openings are indicated by the white arrows. Photo credits: D. Römer.

construction, followed by varying preferences for thick and thin material in the lower part of the turret, and by the selection of thin material for the upper, final part. The placement of single building materials across the turret opening strongly influenced the spatial deposition of additional materials, indicating that building decisions of single workers can impact the final turret architecture.

## 4.1. Behavioural plasticity during turret building

In our experimental series offering artificial sticks, we demonstrated that building activity, i.e. the number of sticks used, was higher at the beginning of the turret rebuilding process, and decreased in each phase. This was not owing to climatic factors, such as daylight or increasing temperatures, as solar radiation and ambient temperature decreased in the initial phase in the evening, and exactly the opposite occurred in the subsequent phase the next morning. Within a building phase, building activity was highest at the beginning and then decreased over the observation time of 1.5 h, in both phases. High building activity at the beginning of turret rebuilding was also shown in a recent study. There, a new turret was assembled in 3 days, yet the interwoven mesh of plant fragments of the turret developed quickly during the first day, while the structure was fully stabilized by cementation with soil pellets during the following 2 days [25]. A nest turret bears a striking similarity to simple buildings engineered by humans. The ants erect a wooden frame supported by beams, interlacing it with thinner, more flexible material to achieve stability, and lastly apply plaster to seal and smooth the structure.

A quick rebuilding process probably aids the protection of the colony by a quick replacement of the lost shelter. The placement of initial sticks across and covering the nest opening, as seen at the beginning of our material choice experiment, could also be a quick response to protect the exposed nest opening after turret removal. A higher initial activity has also been demonstrated for ants when digging [9,32,33]. It appears that both collective building behaviours, excavation and construction, are amplified by self-organization mechanisms, where building activity is rapidly increased at the beginning because of more workers being stimulated to build by responding to cues [33] or by active recruitment to the building site [13]. Which cues *Ac. fracticornis* workers respond to during this phase remains to be investigated. During nest excavation, digging activity then gradually lessens owing to negative feedback as a result of digging activity, i.e. the generation of space and the reduction of worker density at the spot or cavity [10,33]. During turret building, it seems likely that workers obtain feedback information from the structure itself, which leads to a gradual decrease of building activity and finally to the cessation of building.

The preference for building materials also changed over time. Construction was generally started by choosing thick material, followed by a period in which workers showed no preferences for thick and thin

materials, and a final preference for thin material. Workers clearly preferred thick sticks at the initial, and thin sticks at the subsequent building phase. This indicates that workers have actively avoided the collection of thin sticks at the beginning, and of thick sticks at a later phase of turret construction, because the probability to find any type of fragments was the same at the beginning of each building phase. Therefore, workers have the ability to assess the different building materials, although the behavioural mechanisms involved remain unknown.

What are the cues workers could respond to that lead to a change in the selection of the building material used? The observed changes in material choice over time could be based on a simple behavioural rule, i.e. workers may always prefer a thick stick for their first choice when building a new turret from its base, and have no preferences thereafter over the initial building phase. During the subsequent building phase, the observed preferences for thin sticks could be determined by feedback information gained from the turret structure itself, for instance, because of a change in its structural stability, or by changes in local environmental cues (humidity gradients, airflows) during turret growth, leading workers to search for different building materials depending on the actual stage of the building process. The use of different building materials probably aids the stability of the structure, as also seen in the nest wall construction of the ant *T. albipennis* [31]. These ants achieve wall stability by selecting differently sized sand grains when foraging for building material. Even without preferences for building materials, being selective where the materials are incorporated in the built structure is an alternative method to achieve stability, as shown for turret construction in *At. vollenweideri* grass-cutting ants [29]. Showing a temporal change in material preferences in *Ac. fracticornis* might aid turret wall stability by building a thicker, sturdy base out of thick and thin material, and using only the more light and flexible thin material for the upper, thinner part of the conical structure.

Idiothetic cues could also be involved in the temporal shift of material selection. For instance, leaf-cutting ant queens use idiothetic cues to control the depth of their founding nests [34]. By monitoring their movements and comparing it with an internal reference, they switch from excavating a downward leading tunnel to the horizontal excavation of the founding chamber after a certain tunnel length has been reached. Grass-cutting ant workers also appear to use the method of walking up and down a grass blade to determine the length of a grass fragment to cut when foraging [35,36]. Hypothetically, *Ac. fracticornis* workers might walk up and down the turret during construction to indirectly determine its height or to gain feedback from the structure, and alter their material choice when a certain height has been reached and the turret is mechanically stable. It also remains to be explored whether the same workers construct the turret from bottom to top and change their material selection over time, or rather different parts of the turret are built by different workers, each displaying fixed, individual preferences for thick or thin sticks.

## 4.2. Spatial cues for turret construction

After selecting what building material to pick up and to bring to the construction site, workers then have to decide where to place their load. Here, they are guided by local cues from their environment. *Acromyrmex ambiguus* and *Acromyrmex heyeri* use for instance airflows and humidity losses from the nest as a spatial cue where to begin construction [11,12,37] and place leaf fragments around and over nest openings. The outflowing, $CO_2$-enriched air from nest openings also acts as a spatial cue for the deposition of soil pellets when *At. vollenweideri* grass-cutting ants construct their large ventilation turrets [38]. *Acromyrmex fracticornis* workers might also use similar environmental cues emanating from the turret opening to decide whether and where to add to the structure. Our second experimental series with the splitting of the nest entrance demonstrated that worker's decisions where to place a single building material at the beginning of the building process, probably guided by environmental cues, can strongly influence the final architecture of the whole turret, leading to a branched gallery and a turret with multiple openings instead of a turret with a single opening. The placement of a single building material itself could, therefore, act as a quantitative stigmergic cue for the spatial deposition of further material.

Stigmergy is very widespread among social insects to coordinate building. A prominent example is the piling-up of soil pellets to construct pillars in the nest of mound-building termites [15,39]. The deposition of a pellet stimulates additional depositions at the spot by the same or by other workers, thus amplifying the stimulus intensity to place more material there. Leaf-cutting ants also use stigmergic cues during excavation. When digging a nest, soil pellets deposited near the digging site act as stigmergic cues and attract other workers that start excavating there [16], resulting in a collective digging effort at the site. At the very beginning of turret building, *Ac. fracticornis* workers deposit plant fragments tangentially along the edge of the turret opening and sometimes also place sticks across it, a response that may be aimed

to quickly protect the exposed nest entrance, as also described for wood ants at the beginning of nest construction [40]. The nest entrance is kept free over the building process because of the subsequent displacement and removal of those fragments that obstruct the ant traffic. The bodies of passing workers, sometimes also carrying foraged plant material, and the actual traffic flow could act as a physical template for the size of the gallery and turret opening.

Such a simple mechanism could also account for the emergence of branched galleries and thus turrets with several openings. If a single building material is placed across and bisects the nest entrance, and the resulting new openings are large enough as to not obstruct ant traffic, it would most likely not be moved. Then, new material would be either placed or tangentially shifted around the emerging new openings, making the structure grow in height. Quantitative stigmergy, as a mechanism of self-organization through indirect worker interactions, could be coordinating the additional deposition of building material on the initially placed stick, thus increasing the stimulus intensity to trigger further material placement above it. As a consequence, a new gallery wall bisecting the initial opening and resulting in a turret with a branched gallery and two openings would be built via stigmergic responses. Alternatively, the placed material could lead to gallery bifurcation and multiple openings solely by self-organization, without the involvement of stigmergic responses. The stick placed across the opening could, for instance, disrupt worker traffic and lead to the deposition of material at multiple locations. In such a case, the rim of an emergent turret opening, and not the bisecting stick, could act as stigmergic cue for additional material deposition. Whatever the underlying behavioural mechanisms, the architecture of a turret with several openings may have not been selected during evolution as an adaptive response, as it appears to be the case in the grass-cutting ant *At. vollenweideri* in reaction to high $CO_2$ concentrations in the nest [38], but has rather emerged as a by-product of a single building action.

## 4.3. Turrets as an evolutionary adaptation of building behaviour

As building behaviour is energetically costly for the colony, the quick replacement of a destroyed or removed turret highlights the importance of such a structure for colony fitness. *Atta vollenweideri*, the grass-cutting ants well known for building huge nests with turrets that enhance nest ventilation using the Bernoulli principle [22], can also be found near *Ac. fracticornis* nests at the field station where our experiments were performed. By contrast to *Atta* nests, *Ac. fracticornis* nests are rather small and shallow, with the first nest chambers located at 5–10 cm below the surface (F. Roces 2015, personal observations). It seems unlikely that turret construction aids gas exchanges in colonies of this species. The covering of the nest openings by tightly intermeshed plant material could, however, decrease humidity losses of the nest air as shown for other leaf-cutting ant species [12].

Turrets could also aid the prevention of nest floods. Precipitation in the region where *Ac. fracticornis* occurs is characterized by short but heavy rain showers, and areas get temporarily flooded. The turrets of *Ac. landolti*, a phylogenetically closely related grass-cutting ant species [41], can withstand water infiltration for periods of time [28]. An inner lining of the turret with soil pellets, as *Ac. fracticornis* does [25], might increase temporal water impermeability of the turret. We could also observe that during a very rainy field season, their turrets were built with visibly more soil pellets, even on the outside of the turret. The Ponerine ant *Ectatomma opaciventre* also builds turrets that it lines with saliva proteins, probably hardening the structure and aiding water impermeability against flooding [42].

A turret could also prevent dangerous material or organisms such as non-colony members or predators from entering the nest. Many leaf-cutting ant species dispose of their colony waste by external deposits [43] and the waste pile of *Ac. fracticornis* nests is located in very close proximity to the nest entrance. This pathogen-loaded waste material [44] could easily be blown back into a nest opening located at the ground level. Elevating the nest entrance above ground level could protect colony health. In addition, turrets with several openings could aid the organization of the traffic flow of workers going in and out of the nest [45]. A recent study also suggested that the turret could even act as an orientation cue for returning workers [26], although turrets of *Acromyrmex* species are usually inconspicuous and smaller than the surrounding vegetation.

## 5. Conclusion

When referring to 'building' in social insects such as ants and termites, the material is either removed to excavate a structure or piled up to assemble one. By observing material selection and their spatial arrangement during turret building, we could demonstrate that these two seemingly different modes

of construction appear to be organized by the same decentral mechanisms of self-organization, stigmergy, and templates. The observed temporal variation in workers' preferences for building materials adds significantly to the complexity of the built structure, as does the influence of a single building decision on the final architecture as a whole. Plasticity in the use of relatively simple, local building decisions might be one of the reasons for the ecological success of social insects.

Data accessibility. The datasets supporting this article have been uploaded as part of the electronic supplementary material.

Authors' contributions. Conceptualization, D.R., M.I.C. and F.R.; data curation, D.R. and M.I.C.; formal analysis, D.R. and M.I.C.; funding acquisition, D.R. and F.R.; writing—original draft, D.R. and F.R.; writing—review and editing, D.R., M.I.C. and F.R.

Competing interests. The authors declare no conflicts of interest.

Funding. This work was supported by the German Research Foundation (DFG, grant no. SFB 554/TP E1) and the Department of Behavioural Physiology and Sociobiology, Biocenter, University of Würzburg, headed by Prof. Dr Wolfgang Rössler. D.R. was supported by the Postdoc Plus funding programme of the Graduate School of Life Sciences (GSLS), University of Würzburg, Germany, and a Postdoctoral Fellowship of the Agencia Nacional de Investigación e Innovación (ANII, grant no. PD_NAC_2015_1_108641), Uruguay. This publication was supported by the Open Access Publication fund of the University of Wuerzburg.

Acknowledgements. We are very much indebted to the Götz family for providing facilities during the fieldwork in Formosa, Argentina and the ornithologist Alejandro G. Di Giacomo, the station supervisor of the Reserva Ecológica 'El Bagual' (Alparamis SA–Aves Argentinas), and for their friendly support over the years. We also thank Milan Becker for testing in the laboratory the suitability of green pigments to colour the sticks offered as building materials, in the framework of a practical course on behavioural physiology, Griselda Roces for the ant drawing presented in figure 3, and two anonymous reviewers for comments that improved the manuscript. Thanks are also due to Dr Rita Tofalo and Dr Pablo Pazos (Department of Geological Sciences, University of Buenos Aires), and to Dr Daniel Roccatagliata and Dr Brenda Dotti (Department of Biodiversity and Experimental Biology, University of Buenos Aires) for the logistical support of the project.

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
