## [Reviewer comments · Royal Society Open Science]

Review History

RSOS-201312.R0 (Original submission)

Review form: Reviewer 1 (Edith Invernizzi)

Is the manuscript scientifically sound in its present form?

Yes

Are the interpretations and conclusions justified by the results?

No

Is the language acceptable?

Yes

Do you have any ethical concerns with this paper?

No

Have you any concerns about statistical analyses in this paper?

No

Recommendation?

Accept with minor revision (please list in comments)

Comments to the Author(s)

This manuscript looks at nest building by the ant *A. fracticornis*, focusing on the construction of turrets. In particular, the experiments performed look at the use of different building materials in two stages of the building activity and at the effect of one particular environmental conformation (the presence of a tripartition) on the number of entrances in the turret. The experiments presented are meaningful and the authors give a well-thought out statistical analysis of the data. This article represents an advance in our knowledge of insect collective nest building and contains a useful discussion of the potential evolutionary origin of some nest traits. However, the authors interpret some of the findings as an effect of stigmergy, which requires the workers to recognise particular conformations emerging from the building activity and modify their behaviour accordingly. An alternative, simpler explanation (given in the comments) is also possible and the experiments presented do not offer any evidence for stigmergy over alternatives. The distinction is important because 1. the mechanisms and their definitions are well established in the literature and a misuse risks creating confusion in terminology, and 2. different mechanisms have different implications for the evolution of building behaviour (e.g. stigmergy requires that either recognising a similar conformation has had an evolutionary advantage at some point in the past that has led to the evolution of a worker response, or that workers have the cognitive capacity to recognise the conformation as a cue of a potential effect on nest structure and modify their behaviour accordingly). I recommend article acceptance on condition that the authors either modify the article to account for alternative explanations (that should go on an equal level as stigmergy) or, if they have reason to do so, justify their preference of stigmergy as the proposed underlying mechanism and address it explicitly in the text. Other minor suggested edits are listed in the comment file attached (Appendix A).

Review form: Reviewer 2

Is the manuscript scientifically sound in its present form?

Yes

Are the interpretations and conclusions justified by the results?

Yes

Is the language acceptable?

Yes

Do you have any ethical concerns with this paper?

No

Have you any concerns about statistical analyses in this paper?

No

Recommendation?

Accept with minor revision (please list in comments)

Comments to the Author(s)

I think that the study shows clearly that there is a temporal variation in the choice of sticks during the building process. It also presents some clear evidence of an effect of the initial configuration of the structure on the final turret architecture.

Other questions that are also important for understanding the building process remain unanswered by this study: are the same individuals carrying both the thin and the thick sticks? How do ants 'decide' whether to pick a large or a small piece of building material (what are the

stigmergic stimuli determining the subsequent choice of building material)? How do they decide where to deposit the stick (which stigmergic stimuli determine the ant choice)?

Apart from the fact that I would have liked to find an answer to the questions above, my comments are relatively minor and I list them below.

One of the main results, commented at the beginning of the discussion, is that workers showed plasticity in material selection over time. The data provide convincing evidence that the choice of material changes over time, but perhaps it is not possible to conclude that this indicates plasticity as the plasticity of the process was not tested directly (e.g. by removing or adding some sticks to the turret to speed-up or slow down the transition to choosing sticks of a different size). I would suggest considering to talk about temporal variation rather than plasticity (though I agree that plasticity is a likely explanation).

In terms of the result itself that the pick-up choices of workers change over time I do not have particular concerns and I find that the analysis is quite detailed and clear, but I would like to find a very short description of how these pick-up choices were determined. In particular, did it ever happen that a worker picked a stick, but then abandoned it without bringing it to the nest? Or that a worker picked a stick and then brought it inside the nest rather than adding it to the turret at the nest entrance? If these situations did happen, were these considered as valid choices? It is well possible that the choices of ants were always clear, and that all sticks picked by ants were added to the nest entrance, with only very few rare occasions in which the stick was abandoned before being added to the turret, but perhaps it is worth clarifying this for the readers.

A second result, discussed at the beginning of section 4.1, is that building activity decreases over time. Section 4.1 comments that this decrease in activity was not due to climatic factors such as daylight or increasing temperatures. Yet, all the experiments started in the afternoon "because temperatures and sun intensity were lower, and ants were visibly more active". In my comment above about plasticity it was easy to believe that ants exhibit plasticity in their choice of building material, but the experiments themselves do not show it. Similarly for this comment, the fact that building activity decreases over time is easy to believe, but perhaps the experiments themselves do not show this so clearly because you started all the experiments at the same time of the day, when ants "were visibly more active". In the absence of experiments starting at different times, maybe one way to show that the building activity decreased over time would be to compare building activity (number of sticks added per unit time) with some other measure of colony activity at the same time of the day (e.g. foraging). If building activity decreases while general colony activity increases, this would be a good evidence of a decrease over time.

The sticks were marked in order to allow easier recognition. Page 6, line 187 states that unpublished data indicate that grass-cutting ants do not show avoidance or preference for building material coloured with this dye. Could it be worth including these data in the Supplementary Material?

Page 9, line 283: "The sticks were laid across each other at the corners, so that their ends overlapped, like in a human log cabin". This is only shown in the figure, but I wonder if it is possible to give some quantification of the phenomenon and also some description of how it happens: do ants immediately deposit the building material in this configuration, or rather the configuration results from further displacements of the structure (e.g. if a stick blocks the entrance, then it is moved until it naturally takes an orientation tangential to the opening of the nest)?

Page 15, Conclusions section, I find that this section could better reflect the results presented in the manuscript, for instance the first part mentions excavation alongside with piling of material, or the second part comments on the striking similarity to simple buildings engineered by humans. Also the final sentence about ants adding "more flexible material to achieve stability" and "applying plaster to seal and smooth the structure" are only indirectly related to the results of

the study as there was no test of the stability of turrets and the "application of plaster" is not covered in the analyses reported here.

The sentence at page 7, line 201 "although not all colonies showed building activity in either the initial (21 out of the 24 colonies) or the subsequent building phase (19 out of the 24 colonies)" could be rewritten to make it better clear that 21 out of 24 are the colonies that showed building activity.

Page 8, line 230: the pronoun "they" could be ambiguous; I suggest replacing it with "the ants" or similar subject: "the first offered sticks were picked up within a few minutes after they started."

Page 12, line 359: this sentence is slightly difficult to follow. Consider rephrasing it.

Page 14, line 415. Here it is stated that "the architecture of a turret with several openings may have not been selected during evolution as an adaptive response". Was there any suggestion in the previous literature that turrets with multiple openings could have an adaptive value?

Decision letter (RSOS-201312.R0)

Dear Dr Römer,

On behalf of the Editors, we are pleased to inform you that your Manuscript RSOS-201312 "Selection and spatial arrangement of building materials during the construction of nest turrets by grass-cutting ants" has been accepted for publication in Royal Society Open Science subject to minor revision in accordance with the referees' reports. Please find the referees' comments along with any feedback from the Editors below my signature.

Please submit your revised manuscript and required files (see below) no later than 7 days from today's (ie 09-Sep-2020) date. Note: the ScholarOne system will 'lock' if submission of the revision is attempted 7 or more days after the deadline. If you do not think you will be able to meet this deadline please contact the editorial office immediately.

Best regards,
Lianne Parkhouse

Editorial Coordinator
Royal Society Open Science
openscience@royalsociety.org

on behalf of Dr Richard Benton (Associate Editor) and Kevin Padian (Subject Editor)
openscience@royalsociety.org

Reviewer comments to Author:

Reviewer: 1
Comments to the Author(s)

This manuscript looks at nest building by the ant *A. fracticornis*, focusing on the construction of turrets. In particular, the experiments performed look at the use of different building materials in two stages of the building activity and at the effect of one particular environmental conformation (the presence of a tripartition) on the number of entrances in the turret. The experiments presented are meaningful and the authors give a well-thought out statistical analysis of the data. This article represents an advance in our knowledge of insect collective nest building and contains a useful discussion of the potential evolutionary origin of some nest traits. However, the authors interpret some of the findings as an effect of stigmergy, which requires the workers to recognise particular conformations emerging from the building activity and modify their behaviour accordingly. An alternative, simpler explanation (given in the comments) is also possible and the experiments presented do not offer any evidence for stigmergy over alternatives. The distinction is important because 1. the mechanisms and their definitions are well established in the literature and a misuse risks creating confusion in terminology, and 2. different mechanisms have different implications for the evolution of building behaviour (e.g. stigmergy requires that either recognising a similar conformation has had an evolutionary advantage at some point in the past that has led to the evolution of a worker response, or that workers have the cognitive capacity to recognise the conformation as a cue of a potential effect on nest structure and modify their behaviour accordingly). I recommend article acceptance on condition that the authors either modify the article to account for alternative explanations (that should go on an equal level as stigmergy) or, if they have reason to do so, justify their preference of stigmergy as the proposed underlying mechanism and address it explicitly in the text. Other minor suggested edits are listed in the comment file attached.

Reviewer: 2
Comments to the Author(s)

I think that the study shows clearly that there is a temporal variation in the choice of sticks during the building process. It also presents some clear evidence of an effect of the initial configuration of the structure on the final turret architecture.

Other questions that are also important for understanding the building process remain unanswered by this study: are the same individuals carrying both the thin and the thick sticks? How do ants 'decide' whether to pick a large or a small piece of building material (what are the stigmergic stimuli determining the subsequent choice of building material)? How do they decide where to deposit the stick (which stigmergic stimuli determine the ant choice)?

Apart from the fact that I would have liked to find an answer to the questions above, my comments are relatively minor and I list them below.

One of the main results, commented at the beginning of the discussion, is that workers showed plasticity in material selection over time. The data provide convincing evidence that the choice of material changes over time, but perhaps it is not possible to conclude that this indicates plasticity as the plasticity of the process was not tested directly (e.g. by removing or adding some sticks to the turret to speed-up or slow down the transition to choosing sticks of a different size). I would

suggest considering to talk about temporal variation rather than plasticity (though I agree that plasticity is a likely explanation).

In terms of the result itself that the pick-up choices of workers change over time I do not have particular concerns and I find that the analysis is quite detailed and clear, but I would like to find a very short description of how these pick-up choices were determined. In particular, did it ever happen that a worker picked a stick, but then abandoned it without bringing it to the nest? Or that a worker picked a stick and then brought it inside the nest rather than adding it to the turret at the nest entrance? If these situations did happen, were these considered as valid choices? It is well possible that the choices of ants were always clear, and that all sticks picked by ants were added to the nest entrance, with only very few rare occasions in which the stick was abandoned before being added to the turret, but perhaps it is worth clarifying this for the readers.

A second result, discussed at the beginning of section 4.1, is that building activity decreases over time. Section 4.1 comments that this decrease in activity was not due to climatic factors such as daylight or increasing temperatures. Yet, all the experiments started in the afternoon "because temperatures and sun intensity were lower, and ants were visibly more active". In my comment above about plasticity it was easy to believe that ants exhibit plasticity in their choice of building material, but the experiments themselves do not show it. Similarly for this comment, the fact that building activity decreases over time is easy to believe, but perhaps the experiments themselves do not show this so clearly because you started all the experiments at the same time of the day, when ants "were visibly more active". In the absence of experiments starting at different times, maybe one way to show that the building activity decreased over time would be to compare building activity (number of sticks added per unit time) with some other measure of colony activity at the same time of the day (e.g. foraging). If building activity decreases while general colony activity increases, this would be a good evidence of a decrease over time.

The sticks were marked in order to allow easier recognition. Page 6, line 187 states that unpublished data indicate that grass-cutting ants do not show avoidance or preference for building material coloured with this dye. Could it be worth including these data in the Supplementary Material?

Page 9, line 283: "The sticks were laid across each other at the corners, so that their ends overlapped, like in a human log cabin". This is only shown in the figure, but I wonder if it is possible to give some quantification of the phenomenon and also some description of how it happens: do ants immediately deposit the building material in this configuration, or rather the configuration results from further displacements of the structure (e.g. if a stick blocks the entrance, then it is moved until it naturally takes an orientation tangential to the opening of the nest)?

Page 15, Conclusions section, I find that this section could better reflect the results presented in the manuscript, for instance the first part mentions excavation alongside with piling of material, or the second part comments on the striking similarity to simple buildings engineered by humans. Also the final sentence about ants adding "more flexible material to achieve stability" and "applying plaster to seal and smooth the structure" are only indirectly related to the results of the study as there was no test of the stability of turrets and the "application of plaster" is not covered in the analyses reported here.

The sentence at page 7, line 201 "although not all colonies showed building activity in either the initial (21 out of the 24 colonies) or the subsequent building phase (19 out of the 24 colonies)" could be rewritten to make it better clear that 21 out of 24 are the colonies that showed building activity.

Page 8, line 230: the pronoun "they" could be ambiguous; I suggest replacing it with "the ants" or similar subject: "the first offered sticks were picked up within a few minutes after they started."

Page 12, line 359: this sentence is slightly difficult to follow. Consider rephrasing it.

Page 14, line 415. Here it is stated that "the architecture of a turret with several openings may have not been selected during evolution as an adaptive response". Was there any suggestion in the previous literature that turrets with multiple openings could have an adaptive value?

===PREPARING YOUR MANUSCRIPT===

===PREPARING YOUR REVISION IN SCHOLARONE===

- 1) One version identifying all the changes that have been made (for instance, in coloured highlight, in bold text, or tracked changes);
 - 2) A 'clean' version of the new manuscript that incorporates the changes made, but does not highlight them.
 - An individual file of each figure (EPS or print-quality PDF preferred [either format should be produced directly from original creation package], or original software format).
 - An editable file of each table (.doc, .docx, .xls, .xlsx, or .csv).
 - An editable file of all figure and table captions.
- Note: you may upload the figure, table, and caption files in a single Zip folder.
- Any electronic supplementary material (ESM).
 - If you are requesting a discretionary waiver for the article processing charge, the waiver form must be included at this step.
 - If you are providing image files for potential cover images, please upload these at this step, and inform the editorial office you have done so. You must hold the copyright to any image provided.
 - A copy of your point-by-point response to referees and Editors. This will expedite the preparation of your proof.

- Ensure that your data access statement meets the requirements at <https://royalsociety.org/journals/authors/author-guidelines/#data>. You should ensure that you cite the dataset in your reference list. If you have deposited data etc in the Dryad repository, please only include the 'For publication' link at this stage. You should remove the 'For review' link.
- If you are requesting an article processing charge waiver, you must select the relevant waiver option (if requesting a discretionary waiver, the form should have been uploaded at Step 3 'File upload' above).
- If you have uploaded ESM files, please ensure you follow the guidance at <https://royalsociety.org/journals/authors/author-guidelines/#supplementary-material> to include a suitable title and informative caption. An example of appropriate titling and captioning may be found at https://figshare.com/articles/Table_S2_from_Is_there_a_trade-off_between_peak_performance_and_performance_breadth_across_temperatures_for_aerobic_scope_in_teleost_fishes_/3843624.

Author's Response to Decision Letter for (RSOS-201312.R0)

See Appendix B.

Decision letter (RSOS-201312.R1)

Dear Dr Römer,

It is a pleasure to accept your manuscript entitled "Selection and spatial arrangement of building materials during the construction of nest turrets by grass-cutting ants" in its current form for publication in Royal Society Open Science.

on behalf of Prof Kevin Padian (Subject Editor)
openscience@royalsociety.org

Appendix A

This manuscript looks at nest building by the ant *A. fracticornis*, focusing on the construction of turrets. In particular, the experiments performed look at the use of different building materials in two stages of the building activity and at the effect of one particular environmental conformation (the presence of a tripartition) on the number of entrances in the turret. The experiments presented are meaningful and the authors give a well-thought out statistical analysis of the data. This article represents an advance in our knowledge of insect collective nest building and contains a useful discussion of the potential evolutionary origin of some nest traits. However, the authors interpret some of the findings as an effect of stigmergy, which requires the workers to recognise particular conformations emerging from the building activity and modify their behaviour accordingly. An alternative, simpler explanation (given in the comments) is also possible and the experiments presented do not offer any evidence for stigmergy over alternatives. The distinction is important because 1. the mechanisms and their definitions are well established in the literature and a misuse risks creating confusion in terminology, and 2. different mechanisms have different implications for the evolution of building behaviour (e.g. stigmergy requires that either recognising a similar conformation has had an evolutionary advantage at some point in the past that has led to the evolution of a worker response, or that workers have the cognitive capacity to recognise the conformation as a cue of a potential effect on nest structure and modify their behaviour accordingly). I recommend article acceptance on condition that the authors either modify the article to account for alternative explanations (that should go on an equal level as stigmergy) or, if they have reason to do so, justify their preference of stigmergy as the proposed underlying mechanism and address it explicitly in the text. Other minor suggested edits are listed in the comment file attached.

Line 64: influence the collective response directly, through positive or negative feedback loops

68: do not even have to

136-139: There seems to be some confusion regarding the concept of stigmergy. Stigmergy refers to cases where the modification created by a worker's activity is perceived as a cue by all or some workers (see Grassé, 1959; Theraulaz, Bonabeau and Deneubourg, 1998; Camazine *et al.*, 2001). *Perceived* is the keyword here: the worker perceives the modification as a source of information to guide its building and modifies its behaviour as a consequence. It does not include cases where the modification has the *side effect* of creating a change in worker building behaviour.

The behaviour described here could be a side effect of the large fragment disrupting worker flux, which in turn causes the carried material to be deposited at multiple locations. For example, following the authors' description and video, turret construction seems to occur at least partially by workers passing through the inside of the turret, through the galleries already built. Workers might follow directional rules such as "follow the line of nestmates and deposit the material at the edges of the construction". In this case, a disruption caused by an object (the fragment) might lead to a bifurcation in worker flux and thus to multiple entrances being built. I have no knowledge of the process of nest turret construction, beside the information provided with this manuscript, so this concept might not be relevant. This behaviour might well be the effect of stigmergy, if workers are actively responding to information that is conveyed by the branching shape of the stick.

If the authors have specific reasons to rule out self-organisation of building following worker movement as an explanation and prefer stigmergy, then they should outline their reasoning.

Camazine, S. *et al.* (2001) *Self-organization in biological systems*. Princeton University Press.

Grassé, P.-P. (1959) *La Reconstruction du nid et les coordinations interindividuelles chez Bellicositermes*

natalensis et Cubitermes Sp. La théorie de la stigmergie. essai d'interprétation, comportement des termites. Paris: Masson.

Theraulaz, G., Bonabeau, E. and Deneubourg, J.-L. (1998) 'The Origin of Nest Complexity in Social Insects Learning from the models of nest construction', *Complexity*, 3(6).

209-211: individuals (here, the placement of material across the nest opening or turret gallery) influence the final turret architecture

233: materials

254: Despite showing no preference

256: the first choice of stick favoured thick ones

305: I suggest using "base" rather than "beginning", because the preference is only true for the very first step

307-310: I am not clear if this statement (on the use of stigmergic clues) refers to the switch between thick and thin sticks, or to the creation of multiple nest entrances. If it refers to multiple entrances, then see my comment regarding lines 136-139 and also please edit this sentence to clarify.

Regarding the switch between sticks – again, stigmergy is one explanation for what is happening. The analysis did not look at individual level behaviour, so an alternative explanation (and a potential future direction of research) is the existence of a behavioural rule forcing each worker to deposit a thick stick at their first deposition – after which there is no preference for stick type for the base of the turret. This individual-level rule will produce the same pattern in the initial phase as the one observed (*i.e.* an initial peak followed by a gradual decrease in the number of thick sticks used) as more and more builders make their first deposition and carry on building without preference. If this latter hypothesis is true, then the first stick deposited by *each worker* will show a preference, while the following ones will not, independently from whether that worker's activity happens mainly at the very beginning or later on in the process of building the turret's base.

Equally, the transition between absence of a preference and strong preference for thin sticks (in the later phase of building) might be triggered by environmental cues. For example, increased air currents, humidity gradient and change in structural stability (as turret height increases) could be incorporated in the building rules as a trigger for individual workers to change their preference in material. The authors themselves suggest an alternative hypothesis at lines 366-373.

There is no evidence for stigmergy over alternatives, based on the experiments.

389-390: see my comment regarding lines 136-139

412-416: this is self-organisation

429: prevention of nest floods

Supplementary data file: column headings should be ">1 opening" and not "<1 opening"

Appendix B

Response to reviewers

Reviewer: 1

Comments to the Author(s)

This manuscript looks at nest building by the ant *A. fracticornis*, focusing on the construction of turrets. In particular, the experiments performed look at the use of different building materials in two stages of the building activity and at the effect of one particular environmental conformation (the presence of a tripartition) on the number of entrances in the turret. The experiments presented are meaningful and the authors give a well-thought out statistical analysis of the data. This article represents an advance in our knowledge of insect collective nest building and contains a useful discussion of the potential evolutionary origin of some nest traits. However, the authors interpret some of the findings as an effect of stigmergy, which requires the workers to recognise particular conformations emerging from the building activity and modify their behaviour accordingly. An alternative, simpler explanation (given in the comments) is also possible and the experiments presented do not offer any evidence for stigmergy over alternatives. The distinction is important because 1. the mechanisms and their definitions are well established in the literature and a misuse risks creating confusion in terminology, and 2. different mechanisms have different implications for the evolution of building behaviour (e.g. stigmergy requires that either recognising a similar conformation has had an evolutionary advantage at some point in the past that has led to the evolution of a worker response, or that workers have the cognitive capacity to recognise the conformation as a cue of a potential effect on nest structure and modify their behaviour accordingly). I recommend article acceptance on condition that the authors either modify the article to account for alternative explanations (that should go on an equal level as stigmergy) or, if they have reason to do so, justify their preference of stigmergy as the proposed underlying mechanism and address it explicitly in the text.

Answer: *We thank the reviewer for his well made point and we agree. We have included the possible alternative explanation for building outcome at the appropriate points throughout the text (see our detailed comments below). Also, we clarified the mechanism we proposed to be involved, i.e., quantitative stigmergy, where workers would react to the initial deposition of a bisecting stick with the deposition of additional building material on it, thus leading to an increase in the stimulus intensity, which increases the probability that further material would be deposited there.*

Other minor suggested edits are listed in the comment file attached.

Line 64: influence the collective response directly, through positive or negative feedback loops

Answer: *Done, page 2, line 64*

68: do not even have to

Answer: *Done, page 3, line 68*

136-139: There seems to be some confusion regarding the concept of stigmergy. Stigmergy refers to cases where the modification created by a worker's activity is perceived as a cue by all or some workers (see Grassé, 1959; Theraulaz, Bonabeau and Deneubourg, 1998; Camazine *et al.*, 2001). *Perceived* is the keyword here: the worker perceives the modification as a source of information to guide its building and modifies its behaviour as a consequence. It does not include cases where the modification has the *side effect* of creating a change in worker building behaviour.

The behaviour described here could be a side effect of the large fragment disrupting worker flux,

which in turn causes the carried material to be deposited at multiple locations. For example, following the authors' description and video, turret construction seems to occur at least partially by workers passing through the inside of the turret, through the galleries already built. Workers might follow directional rules such as "follow the line of nestmates and deposit the material at the edges of the construction". In this case, a disruption caused by an object (the fragment) might lead to a bifurcation in worker flux and thus to multiple entrances being built. I have no knowledge of the process of nest turret construction, beside the information provided with this manuscript, so this concept might not be relevant. This behaviour might well be the effect of stigmergy, if workers are actively responding to information that is conveyed by the branching shape of the stick.

If the authors have specific reasons to rule out self-organisation of building following worker movement as an explanation and prefer stigmergy, then they should outline their reasoning.

Camazine, S. *et al.* (2001) *Self-organization in biological systems*. Princeton University Press.

Grassé, P.-P. (1959) *La Reconstruction du nid et les coordinations interindividuelles chez *Bellicositermes natalensis* et *Cubitermes Sp.* La théorie de la stigmergie. essai d'interprétation, comportement des termites*. Paris: Masson.

Theraulaz, G., Bonabeau, E. and Deneubourg, J.-L. (1998) 'The Origin of Nest Complexity in Social Insects Learning from the models of nest construction', *Complexity*, 3(6).

Answer: We removed part of the sentence regarding stigmergy from this part of the introduction (page 5, line 137-138). In the discussion, we now included in the text the alternative hypothesis of a purely self-organised behavior leading to the construction of turrets with multiple nest openings, and changed the wording to "quantitative stigmergy" where applicable, as this is the specific stigmergic mechanism we propose could be involved in the process of emergence of turrets with several openings. Page 15, line 434-437

209-211: individuals (here, the placement of material across the nest opening or turret gallery) influence the final turret architecture

Answer: done, page 7, line 213-214

233: materials

Answer: done, page 8, line 236

254: Despite showing no preference

Answer: done, page 9, line 257

256: the first choice of stick favoured thick ones

Answer: done, page 9, line 259

305: I suggest using "base" rather than "beginning", because the preference is only true for the very first step

Answer: done, page 10, line 309

307-310: I am not clear if this statement (on the use of stigmergic clues) refers to the switch between thick and thin sticks, or to the creation of multiple nest entrances. If it refers to multiple entrances,

then see my comment regarding lines 136-139 and also please edit this sentence to clarify.

Answer: *It refers to the latter. We have rewritten the text accordingly to remove stigmergy as sole explanation for the observed behaviour, and took into account the Reviewer's previous comments. Page 11, line 311-314*

Regarding the switch between sticks – again, stigmergy is one explanation for what is happening. The analysis did not look at individual level behaviour, so an alternative explanation (and a potential future direction of research) is the existence of a behavioural rule forcing each worker to deposit a thick stick at their first deposition – after which there is no preference for stick type for the base of the turret. This individual-level rule will produce the same pattern in the initial phase as the one observed (*i.e.* an initial peak followed by a gradual decrease in the number of thick sticks used) as more and more builders make their first deposition and carry on building without preference. If this latter hypothesis is true, then the first stick deposited by *each worker* will show a preference, while the following ones will not, independently from whether that worker's activity happens mainly at the very beginning or later on in the process of building the turret's base.

Answer: *Thanks, again, to the reviewer. This could be a likely explanation for the observed change in workers' preferences over time. We have now included it in the discussion. Page 12, line 357-365.*

Equally, the transition between absence of a preference and strong preference for thin sticks (in the later phase of building) might be triggered by environmental cues. For example, increased air currents, humidity gradient and change in structural stability (as turret height increases) could be incorporated in the building rules as a trigger for individual workers to change their preference in material. The authors themselves suggest an alternative hypothesis at lines 366-373. There is no evidence for stigmergy over alternatives, based on the experiments.

Answer: *Again, thanks! While we do not think that stigmergy is the cause for the observed change in preferences, we now include this alternative explanation in the text. Page 12, line 357-365.*

389-390: see my comment regarding lines 136-139

Answer: *We have now used the term quantitative stigmergy to make clear, to which particular stigmergic mechanism we refer to as one of the possible explanations for the building behaviour observed. Page 14, line 405-406*

412-416: this is self-organisation

Answer: *We have changed the wording in the text accordingly, and clarified that we refer to quantitative stigmergy as a mechanism of self-organised behavior. Page 14, line 428-431*

429: prevention of nest floods

Answer: *done, page 15, line 456*

Supplementary data file: column headings should be ">1 opening" and not "<1 opening"

Answer: *done*

Reviewer: 2

Comments to the Author(s)

I think that the study shows clearly that there is a temporal variation in the choice of sticks during the building process. It also presents some clear evidence of an effect of the initial configuration of the structure on the final turret architecture.

Other questions that are also important for understanding the building process remain unanswered by this study: are the same individuals carrying both the thin and the thick sticks? How do ants 'decide' whether to pick a large or a small piece of building material (what are the stigmergic stimuli determining the subsequent choice of building material)? How do they decide where to deposit the stick (which stigmergic stimuli determine the ant choice)?

Answer: *Very good questions! Unfortunately, so far, we are unable to answer them without further studies in both the field and the laboratory.*

Regarding questions #1: We did not follow individual ants when we made the observations in the field. Material choice over time of individual workers would certainly be worthwhile exploring, for example, by marking builders with an individual color code. We pointed out that it remains to be explored whether the same individuals build the whole turret and change their preferences over time, or different workers with different preferences partake in building at different times during the construction, page 13, line 386-389

Question #2: Again, we cannot answer this question, nor do I think would it be possible based solely on circumstantial field observations. Reviewer #1 hypothesized that it might be an evolved behavioural program of the workers that guides them to prefer a certain type of building material at certain phases of the construction. As probabilistically the ants should encounter both stick types equally (as we offered them in equal numbers), the observed preferences indicate that those ants avoided a stick type while preferring the other (at different building phases), so there seems to be an assessment of building material, although we do not know how. We have now included this point in the discussion, in section 4.1, page 12, line 353-355

Question #3: As reviewer #1 also suggested, we have now also added an alternative hypothesis aside from stigmergic cues (that triggers turret bifurcation) to explain stick placement. It would be certainly worthwhile video-recording the entire building process of turrets, to quantify worker choices, and to develop specific hypothesis that could be tested in new experiments, for instance by offering partially-built turrets and specific sticks at different times of the building process. We think that the study of turret construction in ants can uncover valuable insights into the organization of collective building of social insects. Certainly, a subject we plan to explore further.

Apart from the fact that I would have liked to find an answer to the questions above, my comments are relatively minor and I list them below.

One of the main results, commented at the beginning of the discussion, is that workers showed plasticity in material selection over time. The data provide convincing evidence that the choice of material changes over time, but perhaps it is not possible to conclude that this indicates plasticity as the plasticity of the process was not tested directly (e.g. by removing or adding some sticks to the turret to speed-up or slow down the transition to choosing sticks of a different size). I would suggest considering to talk about temporal variation rather than plasticity (though I agree that plasticity is a likely explanation).

Answer: *We agree. While we think it is rather likely that plasticity is involved, we cannot prove it with the experiments performed. We have therefore adopted the more conservative formulation of temporal variation throughout the text, as suggested.*

In terms of the result itself that the pick-up choices of workers change over time I do not have particular concerns and I find that the analysis is quite detailed and clear, but I would like to find a

very short description of how these pick-up choices were determined. In particular, did it ever happen that a worker picked a stick, but then abandoned it without bringing it to the nest? Or that a worker picked a stick and then brought it inside the nest rather than adding it to the turret at the nest entrance? If these situations did happen, were these considered as valid choices? It is well possible that the choices of ants were always clear, and that all sticks picked by ants were added to the nest entrance, with only very few rare occasions in which the stick was abandoned before being added to the turret, but perhaps it is worth clarifying this for the readers.

Answer: *We did only consider material choices where the picked-up stick was either deposited at or very close to the nest entrance (1-2cm), or, as it happened in a few cases, carried through the nest entrance belowground. The reason for this was that in previous, the material was carried back out of the nest after a short time and added to the pile. The rare occasions in which the offered sticks were lifted but removed further away from the nest entrance, were not considered. We have now added a short explanation to the materials and methods section. Page 7, line 194-196*

A second result, discussed at the beginning of section 4.1, is that building activity decreases over time. Section 4.1 comments that this decrease in activity was not due to climatic factors such as daylight or increasing temperatures. Yet, all the experiments started in the afternoon "because temperatures and sun intensity were lower, and ants were visibly more active". In my comment above about plasticity it was easy to believe that ants exhibit plasticity in their choice of building material, but the experiments themselves do not show it. Similarly for this comment, the fact that building activity decreases over time is easy to believe, but perhaps the experiments themselves do not show this so clearly because you started all the experiments at the same time of the day, when ants "were visibly more active". In the absence of experiments starting at different times, maybe one way to show that the building activity decreased over time would be to compare building activity (number of sticks added per unit time) with some other measure of colony activity at the same time of the day (e.g. foraging). If building activity decreases while general colony activity increases, this would be a good evidence of a decrease over time.

Answer: *This is actually the case (although only based on observations and not quantifications of, for example, foraging activity of a colony). In our experiments, we started observations at two different times of the day, in the evening for the initial phase, and the morning for the subsequent phase. Both times, we could observe a decrease of building activity over time, yet, temperature decreased in the evening (and visible ant activity levels increased), while temperatures increased in the morning (and visible ant activity levels decreased). See page 11, line 318-323 in the discussion. Also, we have slightly rewritten a part of the materials and methods section, to make this point more clear. Page 6, line 166-171*

The sticks were marked in order to allow easier recognition. Page 6, line 187 states that unpublished data indicate that grass-cutting ants do not show avoidance or preference for building material coloured with this dye. Could it be worth including these data in the Supplementary Material?

Answer: *We have now added a graph with these results to the Supplementary material (Figure S1) and included the data in the raw data file.*

Page 9, line 283: "The sticks were laid across each other at the corners, so that their ends overlapped, like in a human log cabin". This is only shown in the figure, but I wonder if it is possible to give some quantification of the phenomenon and also some description of how it happens: do ants immediately deposit the building material in this configuration, or rather the configuration results from further displacements of the structure (e.g. if a stick blocks the entrance, then it is moved until it naturally takes an orientation tangential to the opening of the nest)?

Answer: Unfortunately, we do not have video of the deposition behaviour. Judging by the time lapse video of photos, the answer seems to be both. We observed some sticks that remained at the location where they first appeared in the photos, and also numerous sticks that appear to be displaced, even from one side of the turret entrance to the other. This is likely done by sticks placed in a way that they obstruct the ant traffic, being pushed out of the way simply by workers that were hindered in their movements. We give this explanation in the discussion, page 14, line 418-422.

Page 15, Conclusions section, I find that this section could better reflect the results presented in the manuscript, for instance the first part mentions excavation alongside with piling of material, or the second part comments on the striking similarity to simple buildings engineered by humans. Also the final sentence about ants adding "more flexible material to achieve stability" and "applying plaster to seal and smooth the structure" are only indirectly related to the results of the study as there was no test of the stability of turrets and the "application of plaster" is not covered in the analyses reported here.

Answer: We have moved parts of the conclusion to section 4.1 'Behavioural variation during turret building', page 11, line 327-330, and rewrote part of the conclusion, page 16, line 479-487 to better reflect our findings and their implications for nest architecture of social.

The sentence at page 7, line 201 "although not all colonies showed building activity in either the initial (21 out of the 24 colonies) or the subsequent building phase (19 out of the 24 colonies)" could be rewritten to make it better clear that 21 out of 24 are the colonies that showed building activity.

Answer: done, page 7, line 206-207

Page 8, line 230: the pronoun "they" could be ambiguous; I suggest replacing it with "the ants" or similar subject: "the first offered sticks were picked up within a few minutes after they started."

Answer: We have replaced 'they' with 'the assays', page 8, line 233

Page 12, line 359: this sentence is slightly difficult to follow. Consider rephrasing it.

Answer: done. Also, this sentence has been moved to another paragraph. Page 12, line 356-357

Page 14, line 415. Here it is stated that "the architecture of a turret with several openings may have not been selected during evolution as an adaptive response". Was there any suggestion in the previous literature that turrets with multiple openings could have an adaptive value?

Answer: Yes, Halboth and Roces observed that *Atta vollenweideri*, a species inhabiting clayish and badly ventilated soils, construct nest turrets with multiple openings in reaction to CO₂-rich airflow out of the nest (while constructing turrets with one opening when the outflowing air had low CO₂ levels), presumably because multiple openings increase airflow and therefore help to better ventilate the nest. We have now added this information. Page 15, line 441-443.